# A Critical Role of the IL-22–IL-22 Binding Protein Axis in Hepatocellular Carcinoma

**DOI:** 10.3390/cancers14246019

**Published:** 2022-12-07

**Authors:** Anastasios D. Giannou, Jöran Lücke, Dörte Kleinschmidt, Ahmad Mustafa Shiri, Babett Steglich, Mikolaj Nawrocki, Tao Zhang, Dimitra E. Zazara, Jan Kempski, Lilan Zhao, Olympia Giannou, Theodora Agalioti, Leonie Brockmann, Franziska Bertram, Morsal Sabihi, Marius Böttcher, Florian Ewald, Kornelius Schulze, Johann von Felden, Andres Machicote, Ioannis C. Maroulis, Petra C. Arck, Julia-Kristin Grass, Baris Mercanoglu, Matthias Reeh, Stefan Wolter, Michael Tachezy, Hannes Seese, Myrto Theodorakopoulou, Panagis M. Lykoudis, Asmus Heumann, Faik G. Uzunoglu, Tarik Ghadban, Oliver Mann, Jakob R. Izbicki, Jun Li, Anna Duprée, Nathaniel Melling, Nicola Gagliani, Samuel Huber

**Affiliations:** 1Section of Molecular Immunology und Gastroenterology, I. Department of Medicine, University Medical Center Hamburg-Eppendorf, 20246 Hamburg, Germany; 2Hamburg Center for Translational Immunology (HCTI), University Medical Center Hamburg-Eppendorf, 20246 Hamburg, Germany; 3Department of General, Visceral and Thoracic Surgery, University Medical Center Hamburg-Eppendorf, 20246 Hamburg, Germany; 4Department of Pediatrics, University Medical Center Hamburg-Eppendorf, 20246 Hamburg, Germany; 5Laboratory for Experimental Feto-Maternal Medicine, Department of Obstetrics and Fetal Medicine, University Medical Center Hamburg-Eppendorf, 20246 Hamburg, Germany; 6The Calcium Signaling Group, Department of Biochemistry and Molecular Cell Biology, University Medical Center Hamburg-Eppendorf, 20246 Hamburg, Germany; 7Department of Computer Engineering & Informatics, University of Patras, 26500 Patras, Greece; 8I. Department of Medicine, University Medical Center Hamburg-Eppendorf, 20246 Hamburg, Germany; 9Department of Surgery, University of Patras Medical School, 26500 Patras, Greece; 103rd Department of Surgery, National & Kapodistrian University of Athens, 11527 Athens, Greece; 11Division of Surgery & Interventional Science, University College London (UCL), London NW3 2QG, UK

**Keywords:** hepatocellular carcinoma, IL-22, IL-22BP, T_h_22, neutrophils

## Abstract

**Simple Summary:**

Hepatocellular carcinoma (HCC) still poses a major challenge for curative treatment. Although some new therapeutic options arose during the last decade, the overall prognosis remains poor. New therapies might include the modification of tumor-promoting or -inhibiting mediators of the immune system, such as interleukin (IL)-22 and its natural antagonist IL-22 binding protein (IL-22BP). Thus, this study aimed to investigate the role and underlying mechanisms of IL-22 and IL-22BP signaling in liver cancer. Using two different mouse models, we found that IL-22 promoted HCC development, while IL-22BP led to reduced tumor growth. IL-22 was mainly produced by a subset of T cells in HCC, whereas IL-22BP was abundantly secreted by neutrophils. Importantly, we identified hepatocytes as a major target of this pathological IL-22-signaling. Moreover, abrogation of IL-22 signaling in hepatocytes reduced *STEAP4* expression-a known oncogene-in an HCC mouse model in vivo, and *STEAP4* expression correlated with *IL22* levels in human HCC samples. Taken together, these data might pave the way for new therapeutical approaches by blocking IL-22 or its downstream signaling in HCC.

**Abstract:**

Hepatocellular carcinoma (HCC) ranks among the five most common cancer entities worldwide and leads to hundred-thousands of deaths every year. Despite some groundbreaking therapeutical revelations during the last years, the overall prognosis remains poor. Although the immune system fights malignant transformations with a robust anti-tumor response, certain immune mediators have also been shown to promote cancer development. For example, interleukin (IL)-22 has been associated with HCC progression and worsened prognosis in multiple studies. However, the underlying mechanisms of the pathological role of IL-22-signaling as well as the role of its natural antagonist IL-22 binding protein (IL-22BP) in HCC remain elusive. Here, we corroborate the pathogenic role of IL-22 in HCC by taking advantage of two mouse models. Moreover, we observed a protective role of IL-22BP during liver carcinogenesis. While IL-22 was mainly produced by CD4^+^ T cells in HCC, IL-22BP was abundantly expressed by neutrophils during liver carcinogenesis. Hepatocytes could be identified as a major target of this pathological IL-22-signaling. Moreover, abrogation of IL-22 signaling in hepatocytes in *IL22ra1^flox/flox^* × *Alb^Cre+^* mice reduced *STEAP4* expression-a known oncogene-in HCC in vivo. Likewise, *STEAP4* expression correlated with *IL22* levels in human HCC samples, but not in healthy liver specimens. In conclusion, these data encourage the development of therapeutical approaches that target the IL-22–IL-22BP axis in HCC.

## 1. Introduction

Hepatocellular carcinoma (HCC) is the most common primary tumor of the liver and is often associated with a pre-damaged and cirrhotic liver [1]. It ranks among the five most common cancer entities worldwide and causes hundred-thousands of cancer-related deaths every year [2]. Many patients in Europe and Northern America that suffer from HCC are diagnosed at a rather late time point [3], rendering them unsuitable for potentially curating therapies such as surgical resection, radiofrequency ablation (RFA), or liver transplantation [4]. Although some progress has been made during the last years in treating even advanced stages of HCC by introducing multikinase inhibitors and checkpoint inhibitors to the therapeutic regimes [5,6], the overall prognosis remains poor. In Europe, around 80% of patients die within 5 years of the initial diagnosis of HCC [7], justifying the urgent need for innovative therapies to limit tumor progression.

Among potential therapeutic options, the inhibition of tumor-promoting aspects of the immune system plays more and more a central role. While the immune system and its mediators foremost contribute to a potent anti-tumor response, certain mechanisms are indeed capable of promoting cancer under certain circumstances and thus, might have an overall detrimental effect on the host [8]. With regard to the cytokine interleukin (IL)-22, dichotomous functions with partial tumor-promoting effects have been described in many studies [9,10].

IL-22 is a mainly pro-inflammatory cytokine that maintains homeostatic effects at barrier sites through its anti-infective and genome-protective effects [11]. It is produced by a variety of immune cells, among them natural killer T (NKT) cells [12], gamma delta (γδ) T cells [10,13,14], T helper 17 (T_h_17) cells [14,15,16], T helper 22 (T_h_22) cells [17] or type 3 innate lymphoid cells (ILC3s) [10,18]. IL-22 signals through a heterodimeric receptor consisting of the ubiquitously expressed IL-10 receptor 2 (IL-10R2) and the IL-22-specific receptor IL-22 receptor 1 (IL-22R1) [19,20], which is predominately expressed on non-hematopoietic cells [21]. A second soluble receptor can equally bind to IL-22 and, thus, is termed IL-22 binding protein (BP) [9]. IL-22BP can therefore neutralize the bioactivity of IL-22 since it inhibits the binding to its membrane-bound counterpart [9,22,23,24]. Thus, a delicate balance between IL-22 and IL-22BP is generally needed physiologically, since both the lack of IL-22 as well as its uncontrolled expression can lead to pathogenesis such as infections [25] or cancer development [9,26], respectively.

In the liver, IL-22 is capable of protecting from both invasion of pathogens, as well as tissue harm through toxic damage [27]. Furthermore, IL-22 can enhance liver regeneration by reducing the expression of apoptotic genes and increasing the expression of proliferative genes in hepatocytes [28]. In addition, IL-22 was identified to possess anti-tumorigenic traits in different tumor models of colorectal and breast cancer [9,10,29]. However, as mentioned above, many studies also reported a robust pathogenic effect of IL-22 in different preclinical mouse models [9,15,30,31,32,33,34,35,36,37,38,39,40], although mechanistic explanations for this dichotomous effect of IL-22 remain scarce.

The first studies investigated the role of IL-22 during the development of HCC as early as 2011 [41,42]. By using IL-22-deficient mice [42] as well as IL-22 transgenic (TG) mice [41], which continuously overexpress this cytokine, an HCC-promoting effect of IL-22 was concluded. In line with these observations, an increased number of IL-22-producing cells infiltrating the tumor and high levels of IL-22 in the serum of patients were found to be negative prognostic factors for the progression of HCC [43,44,45]. Nonetheless, specific target cells that might explain the pathogenic influence of IL-22 during HCC development remain yet to be elucidated. Moreover, the relevancy of IL-22BP during liver carcinogenesis has not been studied so far.

To answer these questions, we investigated the effect of IL-22 and IL-22BP using two different murine models of HCC development. Indeed, deficiency of IL-22 protected from HCC development. Furthermore, we are the first ones to show a significant role of endogenous IL-22BP in controlling IL-22 in liver cancer. IL-22 was mainly produced by CD4^+^ T cells during liver carcinogenesis, while IL-22BP was abundantly secreted by neutrophils during HCC development. Mechanistically, we found that IL-22 exerted these pathogenic effects by acting directly on hepatocytes. Furthermore, RNA sequencing of hepatocytes revealed five genes that were upregulated upon IL-22 stimulation in vitro, many of them known to possess tumor-promoting effects in similar contexts. Indeed, many of these genes are also correlated with *IL22* expression in the liver of patients suffering from HCC. Strikingly, a correlation in non-tumorous livers could not be determined. Taken together, our data indicate a critical role of the IL-22–IL-22BP axis in HCC and we revealed potential downstream targets of IL-22 signaling.

## 2. Materials and Methods

### 2.1. Human Samples and Studies

Liver biopsies were taken from patients suffering from HCC or patients suffering from steatohepatitis without any signs of malignant transformation. Human studies were approved by the local ethical committee “Ethik-Kommission der Ärztekammer Hamburg” under the approval codes PV-3578 and PV-3548. Written informed consent to the study protocol was obtained from all participants before inclusion in this study.

### 2.2. Animals

C57BL/6J, *IL22^-/-^*, *IL22bp^-/-^*, *Il10^eGFP^* × *Foxp3^mRFP^* × *Il17a^Katushka^* × *IL22^sgBFP^*, *IL22ra1^-/-^*, *IL22ra1^flox/flox^* × *Alb^Cre+^* and *IL22ra1^flox/flox^* × *Cdh5^Cre+^* mice were bred and housed under specific pathogen-free conditions in the animal facility of the University Medical Center Hamburg Eppendorf. Age- and sex-matched littermates were used. All experiments were carried out in accordance with the Institutional Review Board “Behörde für Justiz und Verbraucherschutz (Veterinärwesen/Lebensmittelsicherheit)” (Hamburg, Germany).

### 2.3. Chemical Induction of Liver Carcinogenesis

The induction of HCC via the DEN-TCPOBOP approach is well described [46,47]. In brief, fourteen-day-old mice were intraperitoneally (i.p.) injected with diethylnitrosamine (DEN) once (20 mg/kg body weight, dissolved in saline). From the fourth week of life, these mice received TCPOBOB (3 mg/kg body weight, dissolved in corn oil) i.p. for tumor promotion every two weeks. At six months of age, mice were sacrificed. The tumor burden was assessed by macroscopic and microscopic counting as well as by determining the liver weight.

### 2.4. Choline-Deficient High-Fat Diet

This model was carried out according to previous descriptions [48]. Fourteen-day-old mice were injected with DEN i.p. once (20 mg/kg body weight, in saline). From the fourth week of life, these mice were constantly fed a choline-deficient high-fat diet. The usual chow diet was used as a control diet. At six months of age, mice were sacrificed. The tumor burden was assessed by macroscopic and microscopic counting as well as by determining the liver weight.

### 2.5. Magnetic Resonance Imaging (MRI)

The development of HCC was monitored by MRI (Bruker MRI) examinations once a month starting at three months of age. During the MRI examinations, the mice were narcotized with isoflurane by inhalation. Based on the MRI, the tumor volume was determined.

### 2.6. Measurement of Alanine Transaminase (ALT)

Plasma samples were diluted and ALT enzyme levels were analyzed at the Department of Clinical Chemistry (University Medical Center Hamburg- Eppendorf, Hamburg, Germany).

### 2.7. Leucocyte Isolation from the Liver

Leucocytes were isolated from the murine liver with or without HCC. First, mice were euthanized and their liver was perfused with PBS via the inferior vena cava and the portal vein. After the removal of the gall bladder, the liver was excised. Then, the murine tissue was cut into small pieces and minced using scissors. Subsequently, the hepatic tissue was incubated for 30 min at 37 °C on a shaking incubator in HBSS (including Ca^2+^ and Mg^2+^), supplemented with Collagenase (1 mg/mL) and DNase I (10 U/mL). After a washing step with PBS and 1% FBS, leucocytes were further enriched by a Percoll gradient (GE Healthcare, Chicago, IL, USA).

### 2.8. Fluorescent-Activated Cell Sorting (FACS)

Fc-γ receptors were blocked using a monoclonal antibody (clone 2.4G2). The cells were stained with fluorochrome-conjugated antibodies (Appendix A). BD LSRFortessa and FACSAria (BD Biosciences, San Jose, CA, USA) were used for cell analysis and cell sorting, respectively. Data were analyzed using FlowJo v.6.1 (Tree Star, Ashland, OR, USA).

### 2.9. Isolation of Primary Hepatocytes

The isolation was carried out according to standard protocols [49]. First, the liver was digested by perfusion with Liberase (Roche Diagnostics, Basel, Switzerland) and was then gently disrupted to free residual cells. The single-cell suspension was filtered through a 100 μm cell strainer and the cells were allowed to settle by gravity for 20 min. Subsequently, parenchymal cells were separated by 10 min centrifugation in a 90% Percoll gradient (GE Healthcare, Chicago, IL, USA). For primary hepatocyte culture, William’s E + GlutaMAX -I medium (Life Technologies, Karlsruhe, Germany) was supplemented with 10% FBS (Life Technologies, Karlsruhe, Germany), 1% penicillin/streptomycin (Life Technologies, Karlsruhe, Germany), and 1% L-glutamine (Life Technologies, Karlsruhe, Germany). Cells were incubated overnight at 37 °C with 40% O_2_. On the next day, the hepatocytes were washed and incubated with 1 ng/mL recombinant mIL-22 (eBioscience, San Diego, CA, USA) for 15 min.

### 2.10. Murine RNA Extraction

Total RNA was extracted from tissue or isolated cells as indicated in the main text. A standard protocol using TRIzol^®^ Reagent (Invitrogen, Waltham, MA, USA) was used.

### 2.11. Human RNA Extraction

Total RNA was extracted from tissue using the Rneasy^®^ Plus Mini Kit (Qiagen, Hilden, Germany) according to the manufacturer’s instructions.

### 2.12. cDNA Synthesis and qPCR

The High-capacity cDNA Synthesis Kit (Applied Biosystems, Waltham, MA, USA) was used for cDNA synthesis. Probes were purchased from Applied Biosystems (Appendix A). Real-time PCR was performed using the Kapa Probe Fast qPCR Master Mix (Kapa Biosystems, Kapa Biosystems) on the StepOne Plus system (Applied Biosystems, Waltham, MA, USA). For both humans and mice, relative expression was normalized to HPRT and calculated using the 2-ΔΔCt method.

### 2.13. RNA-seq

Using 2 mg of RNA per sample, sequencing libraries were generated using NEBNext UltraTM RNA Library Prep Kit for Illumina (New England Biolabs, Ipswich, MA, USA). cDNA libraries were sequenced on Illumina HiSEquation 2500 yielding ∼15 million 50 bp single-end reads per sample. Overall quality was assessed with FastQC v. 0.11.5, low-quality bases were trimmed off with Trimmomatic v. 0.33 [50], followed by alignment to the Mus musculus genome draft GRCm38.84 using STAR v. 2.5.0 [51]. For visualization and hierarchical clustering, reads were normalized using the transcripts per million method [52], but raw read counts were used for differential expression analysis using DESeq2 v. 1.14 [53].

### 2.14. Statistical Analysis

Sample size was calculated using G*power, assuming α = 0.05, β = 0.8, and ρ = 0.3. No data were excluded. Transgenic animals were enrolled case–control-wise and data acquisition was blinded. Statistical analysis was performed using GraphPad Prism^®^ Software (GraphPad Software, San Diego, CA, USA). For paired group comparison, the non-parametric two-sided Mann–Whitney test was used. Sample size (*n*) refers to biological replicates. The mRNA expression of the cytokines was transformed using a base 2 logarithm. The significance level α was set to 0.05.

## 3. Results

### 3.1. A Critical Role of the IL-22–IL-22BP Axis in HCC Mouse Models

To investigate the effects of the cytokine IL-22 and its natural inhibitor IL-22BP during HCC development, we applied a well-known chemical model of HCC induction to C57BL/6-, *IL22^-/-^*-, and *IL22bp^-/-^*-mice. Fourteen-day-old mice were injected once with DEN. From the fourth week of life, these mice were then injected with TCPOBOP every two weeks for six months (Figure 1A). When assessing the tumor burden in six-month-old mice, *IL22*-deficient mice indeed possessed significantly less tumor burden than the wild-type control ones, while *IL22bp*-deficient mice had significantly more tumors than C57BL/6 mice (Figure 1B). An equal observation could be made by performing MR-imaging of the livers of these mice that allowed a three-dimensional assessment of the tumor volume (Figure 1C). In line with these results, the liver weight and transaminase levels–which allow an assessment of liver damage–were also elevated in *IL22bp*-deficient animals compared to *IL22*-deficient mice (Figure 1D,E). Taken together, we could demonstrate that IL-22 promotes HCC development, while IL-22BP attenuates tumorigenesis in the liver.

To corroborate our findings, we applied a second model for HCC development during which we fed the mice with a special high-fat diet after an initial injection with DEN (Figure 2A). Indeed, this model reproduced the initial findings, since *IL22*-deficient mice were once again protected from tumor development, while *IL22bp*-deficient mice developed significantly more tumors than the control mice (Figure 2B). Using MR-imaging, we likewise found a reduction in tumor volume in *IL22^-/-^*-mice, while *IL22bp^-/-^*-mice displayed an increased tumor volume (Figure 2C). Moreover, *IL22bp*-deficient animals also displayed increased liver weight and transaminase levels compared to *IL22^-/-^*-mice (Figure 2D,E). Taken together, IL-22 displays pathogenic effects in the two examined murine models of HCC, while IL-22BP exerts significant protective effects in both the chemical and high-fat-diet model of liver tumor development. 

### 3.2. CD4^+^ T Cells and Neutrophils Show a High Expression of IL-22 and IL-22BP in HCC, Respectively

We next wanted to determine the cellular sources of IL-22 and IL-22BP during HCC development, respectively. To that end, we chemically induced liver tumors with DEN/TCPOBOP treatment in reporter mice, in which IL-10-producing cells co-express green fluorescent protein (GFP), Foxp3-positive cells red fluorescent protein (RFP), IL-17A-producing cells Katushka and IL-22-producing cells blue fluorescent protein (BFP) (Figure 3A). When then performing an unsupervised t-SNE analysis, we found that IL-22 production was mainly concentrated in two clusters, with one of them co-expressing IL-17A (Appendix A and Figure 3B). Further analysis revealed that these clusters consisted of T_h_17 cells (defined as CD45^+^ CD3^+^ CD4^+^ Foxp3^-^ IL-17A^+^ IL-22^+^) and T_h_22 cells (defined as CD45^+^ CD3^+^ CD4^+^ Foxp3^-^ IL-17A^-^ IL-22^+^), respectively (Figure 3C). Indeed, both frequencies of IL-22-producing T_h_17 and T_h_22 cells increased in HCC compared to healthy controls (Figure 3C). In line with this, CD4^+^ T cells showed the highest *IL22* expression compared to the other analyzed cell types in HCC (Figure 3D). To identify the source of IL-22BP in HCC, we sorted different immune cell populations and measured the levels of *IL22bp* expression with qPCR. We found that both CD4^+^ T cells (defined as CD45^+^ CD3^+^ CD4^+^) as well as CD8^+^ T cells (defined as CD45^+^ CD3^+^ CD8^+^) showed increased *IL22bp* expression in HCC compared to control (Figure 3E). Nonetheless, the highest *IL22bp* expression both in the healthy liver as well as in the liver with liver tumors was found in neutrophils (defined as CD45^+^ CD11b^+^ Ly6G^+^) (Figure 3E). In summary, we found that CD4^+^ T cells, and in particular T_h_22 cells show a high expression of IL-22 during HCC, while neutrophils, besides other cells, show a high expression of IL-22BP during liver carcinogenesis. 

### 3.3. IL-22 Signaling in Hepatocytes Promotes HCC Development

In the next step, we sought to investigate possible mechanisms and cellular targets, through which IL-22 promotes tumor development. Thus, we chemically induced HCC in different mouse lines (Figure 4A). As expected, mice lacking the receptor for IL-22, IL-22RA1, were protected from tumor development (Figure 4B). To narrow down potential targets, we used conditional mouse lines, in which IL-22RA1 is only depleted on specific cell subsets. Since IL-22RA1 is primarily expressed on non-hemopoietic cells, we focused our investigation on endothelial cells and hepatocytes. To that end, *IL22ra1^flox/flox^*; *Cdh5^Cre+^* mice were used to specifically deplete *IL22ra1* on endothelial cells. Interestingly, these mice did not develop fewer liver tumors than *IL22ra1^+/+^*; *Cdh5^Cre+^* control mice (Figure 4C). Thus, IL-22 does not act on endothelial cells to promote hepatocellular cancer.

Next, we chemically induced HCC in *IL22ra1^flox/flox^*; *Alb^Cre+^* mice, which lack the expression of *IL22ra1* on hepatocytes. Strikingly, mice lacking *IL22ra1* expression on hepatocytes were protected from HCC development (Figure 4D). Taken together, IL-22 exerts its tumor-promoting effects mainly via IL-22RA1-mediated signaling on hepatocytes.

### 3.4. Deficiency of IL-22 Signaling in Hepatocytes Leads to Downregulation of STEAP4 in HCC

To elucidate potential downstream mechanisms that are induced by IL-22 signaling within hepatocytes, we isolated hepatocytes from murine livers and stimulated them in vitro with recombinant IL-22 (rIL-22) (Figure 5A). After performing RNA sequencing, we identified several upregulated genes in hepatocytes upon rIL-22 stimulation, including *STEAP4*, *IL33*, *FGA*, *FGB*, and *CEBPD* (Figure 5B,C). Indeed, validation of these results with qPCR revealed that these five genes were drastically upregulated in hepatocytes upon stimulation with IL-22 (Figure 5D). To verify that IL-22 also regulated these genes within hepatocytes and during carcinogenesis in vivo, we measured their RNA levels in *IL22ra1^flox/flox^*; *Alb^Cre+^* mice as well as their matched controls after HCC development. Indeed, *IL22ra1^flox/flox^*; *Alb^Cre+^* mice displayed a significantly reduced expression of *STEAP4* and *FGA*, while expression levels of the other three genes remained unchanged (Figure 5E). As a control, we also measured the expression levels of these five genes in *IL22ra1^flox/flox^*; *Cdh5^Cre+^* mice as well as their matched controls. As expected, no difference in gene expression levels in any of the five target genes of IL-22 could be detected (Figure 5F).

### 3.5. STEAP4 Correlates with IL22 in Patient Samples of HCC

Finally, we investigated the expression of these five IL-22-associated genes in human samples of HCC and appropriate controls. Indeed, *STEAP4*, *CEBPD*, and *IL33* were positively correlated to *IL22* expression levels in human HCC, but not in healthy liver specimens (Figure 6A–C). Such a correlation could neither be found for *FGA* in human HCC (Figure 6D) nor for *FGB* (Figure 6E). In summary, *IL22* expression correlates with *STEAP4* in samples of HCC, but not in healthy liver specimens. Likewise, a correlation of *CEBPD* or *IL33* with *IL22* could be equally detected.

## 4. Discussion

In this study, we investigated the roles of IL-22 and its endogenous inhibitor, IL-22BP, during the development of HCC. Using two different mouse models, we found a pathogenic role for IL-22 but showed that IL-22BP exerts protective effects in the same setting. We determined CD4^+^ T cells and neutrophils as significant sources for IL-22 and IL-22BP during HCC, respectively. Furthermore, we showed that IL-22 acted on hepatocytes by increasing the expression of different pro-tumorigenic genes, such as *STEAP4* (Figure 6F). Moreover, a correlation between *STEAP4* and IL-22 could also be shown in human HCC liver samples.

The immune system possesses potent anti-tumor functions that inhibit tumor formation in many settings. However, tumors can also exploit certain parts of the host’s immune response to promote cancer growth, often by subverting cues for wound healing. Thus, it is no wonder that IL-22, a cytokine that promotes tissue regeneration in many settings, can also promote tumors. Regarding the development of HCC, two landmark studies have previously explored the influence of IL-22. Using a transgenic mouse strain, in which IL-22 was artificially overexpressed, a pathogenic effect on HCC development upon DEN injection was observed [41]. Likewise, a reduced HCC count upon DEN treatment in *IL22*-deficient mice could be observed in a second study [42]. Our study corroborates these findings by adding two further murine HCC models to the record, which show that *IL22* deficiency indeed protects from tumor development in the liver.

Moreover, we demonstrate for the first time that IL-22BP protects from liver tumor development and that IL-22BP is produced from neutrophils in addition to dendritic cells (DCs), CD4^+^, and CD8^+^ T cells during HCC development. The anti-tumorigenic effect of IL-22BP is well documented in colorectal cancer, while its effect on other tumor entities is not known to this current date. In colorectal cancer, the source of IL-22BP was determined to consist of DCs, CD4^+^ T cells, and eosinophils. Our study identifies neutrophils as an additional potential source of IL-22BP. We additionally sought to determine the cellular source of IL-22 in HCC. In line with former publications [45,54], we found that both T_h_17 and T_h_22 cells can produce IL-22 in steady state and liver tumors. 

Many studies imply that IL-22 exerts its tumor-promoting effects directly by inducing a STAT3-mediated signal cascade in the malignantly transformed hepatocytes [42,45,55]. However, corresponding in vivo examinations of these and other cellular targets were missing. Here, we identified hepatocytes as the main cellular target of IL-22 during liver carcinogenesis in vivo and showed that signaling of this cytokine on endothelial cells is dispensable for overall tumor development.

Furthermore, we identified potential downstream targets of IL-22-among them *STEAP4*. Indeed, abrogation of IL-22 signaling in hepatocytes in *IL22ra1^flox/flox^* × *Alb^Cre+^* mice reduced *STEAP4* in HCC in vivo. Finally, *IL22* expression also correlated with *STEAP4* in samples of HCC in humans, but not in healthy liver specimens. *STEAP4* was previously shown to enhance tumorigenesis and metastasis in the colon by fueling copper metabolism [56,57]. Equivalent to our finding, *STEAP4* was upregulated by IL-22 in this study, but also by other cytokines such as IL-17A. However, another study describes this gene as a tumor-suppressing gene whose expression is downregulated in HCC [58]. Thus, further studies are warranted to clarify the role of *STEAP4* in HCC.

## 5. Conclusions

In summary, this study highlights the pro-tumorigenic effect of IL-22 in liver carcinogenesis while underlining the anti-tumorigenic effects of IL-22BP. Further analysis suggests that IL-22 acts directly on hepatocytes thereby promoting tumorigenesis. Thus, the IL-22–IL-22BP axis is a novel target in HCC.

## Figures and Tables

**Figure 1 cancers-14-06019-f001:**
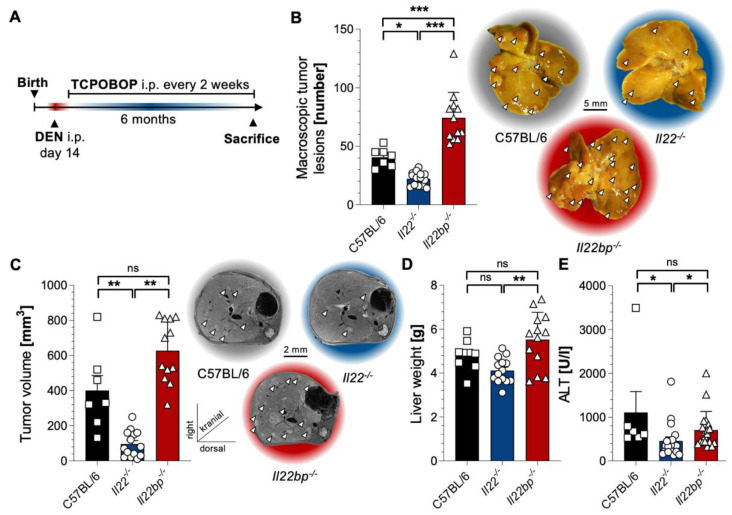
IL-22 and IL-22BP display contrary effects in a chemical mouse model of HCC. (**A**) A schematic timeline describing chemical HCC induction with DEN and TCPOBOP in mice; (**B**) left: number of macroscopic tumor lesions in livers of 6 -month-old mice that were either wild type (C57BL/6, left, black, *n* = 7), *IL22*-deficient (*IL22^-/-^*, middle, blue, *n* = 14) or *IL22bp*-deficient (*IL22bp^-/-^*, right, red, *n* = 12); right: representative macroscopic pictures; (**C**) left: tumor volume (in mm^3^) assessed by MRI in livers of 6 month-old mice after HCC induction that were either wild type (C57BL/6, left, black, *n* = 7), *IL22*-deficient (*IL22^-/-^*, middle, blue, *n* = 14) or *IL22bp*-deficient (*IL22bp^-/-^*, right, red, *n* = 12); right: representative macroscopic pictures; (**D**) liver weight of 6 -month-old mice after HCC induction that were either wild type (C57BL/6, left, black), *IL22*-deficient (*IL22^-/-^*, middle, blue) or *IL22bp*-deficient (*IL22bp^-/-^*, right, red); (**E**) serum ALT levels (in U/l) of 6-month-old mice after HCC induction that were either wild type (C57BL/6, left, black), *IL22*-deficient (*IL22^-/-^*, middle, blue) or *IL22bp*-deficient (*IL22bp^-/-^*, right, red). Data are pooled from 2 independent experiments. Data presented as mean ± SEM. ns: *p* > 0.05; *: *p* < 0.05; **: *p* ≤ 0.01; ***: *p* ≤ 0.001 as assessed by Mann–Whitney U test.

**Figure 2 cancers-14-06019-f002:**
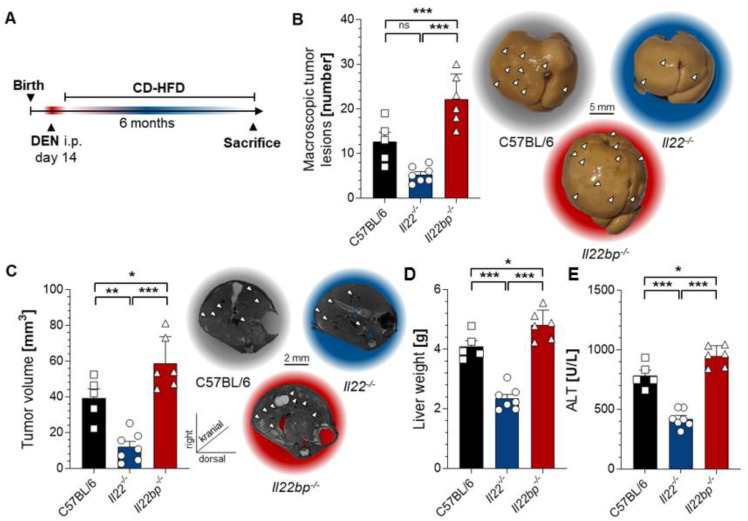
IL-22 and IL-22BP display contrary effects in a nutrition-based mouse model for HCC. (**A**) A schematic timeline describing the used nutrition-based HCC induction with CD-HFD in mice; (**B**) left: number of macroscopic tumor lesions in livers of 6 -month-old mice that were either wild type (C57BL/6, left, black, *n* = 5), *IL22*-deficient (*IL22^-/-^*, middle, blue, *n* = 7) or *IL22bp*-deficient (*IL22bp^-/-^*, right, red, *n* = 6); right: representative macroscopic pictures; (**C**) left: tumor volume (in mm^3^) assessed by MRI in livers of month-old mice after CD-HFD feeding that were either wild type (C57BL/6, left, black, *n* = 5), *IL22*-deficient (*IL22^-/-^*, middle, blue, *n* = 7) or *IL22bp*-deficient (*IL22bp^-/-^*, right, red, *n* = 6); right: representative macroscopic pictures; (**D**) liver weight of 6 -month-old mice after CD-HFD feeding that were either wild type (C57BL/6, left, black), *IL22*-deficient (*IL22^-/-^*, middle, blue) or *IL22bp*-deficient (*IL22bp^-/-^*, right, red); (**E**) serum ALT levels (in U/l) of 6-month-old mice after CD-HFD feeding that were either wild type (C57BL/6, left, black), *IL22*-deficient (*IL22^-/-^*, middle, blue) or *IL22bp*-deficient (*IL22bp^-/-^*, right, red). Data are pooled from 2 independent experiments. Data presented as mean ± SEM. ns: *p* > 0.05; *: *p* < 0.05; **: *p* ≤ 0.01; ***: *p* ≤ 0.001 as assessed by Mann–Whitney U test.

**Figure 3 cancers-14-06019-f003:**
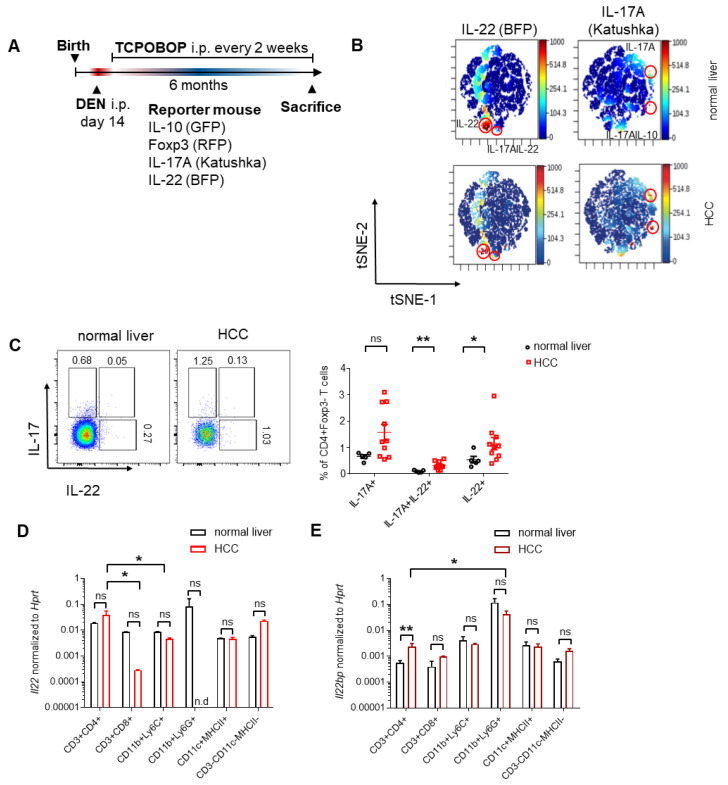
T_h_22 cells and neutrophils comprise the major cellular sources of IL-22 and IL-22BP in HCC, respectively. (**A**) A schematic timeline describing chemical HCC induction with DEN and TCPOBOP in mice; (**B**) unbiased tSNE-analysis of CD45^+^ leucocytes analyzed with FACS isolated from the healthy murine liver (upper panel, *n* = 5) or murine liver after chemical induction of HCC (lower panel, *n* = 5), color change indicates levels of IL-22 (left panels) or IL-17A (right panels); (**C**) left: representative FACS dot plots of CD45^+^ CD3^+^ CD4^+^ Foxp3^-^ leucocytes analyzed with FACS isolated from the healthy murine liver (left) or murine liver after chemical induction of HCC (right); right: percentage of IL-17A^+^IL-22^-^ (left), IL-17A^+^IL-22^+^ (middle) and IL-17A^-^IL-22^+^ (right) CD45^+^ CD3^+^ CD4^+^ Foxp3^-^ leucocytes analyzed with FACS isolated from the healthy murine liver (black, *n* = 5) or murine liver after chemical induction of HCC (red, *n* = 5); (**D**) relative expression of *IL22* in comparison to *HPRT* of indicated leucocyte subsets isolated from either healthy murine liver (black, *n* = 3 pooled samples of sorted cells, for each pooled sample 4 mice were used) or murine liver after chemical induction of HCC (red, *n* = 3 pooled samples of sorted cells, for each pooled sample 4 mice were used). (**E**) relative expression of *IL22bp* in comparison to *HPRT* of indicated leucocyte subsets isolated from either healthy murine liver (black, *n* = 12) or murine liver after chemical induction of HCC (red, *n* = 12). Leucocytes were all cell-sorted on CD45^+^ and then on indicated markers. Data are pooled from 2 independent experiments. Data presented as mean ± SEM. ns: *p* > 0.05; *: *p* < 0.05; **: *p* ≤ 0.01 as assessed by Mann–Whitney U test.

**Figure 4 cancers-14-06019-f004:**
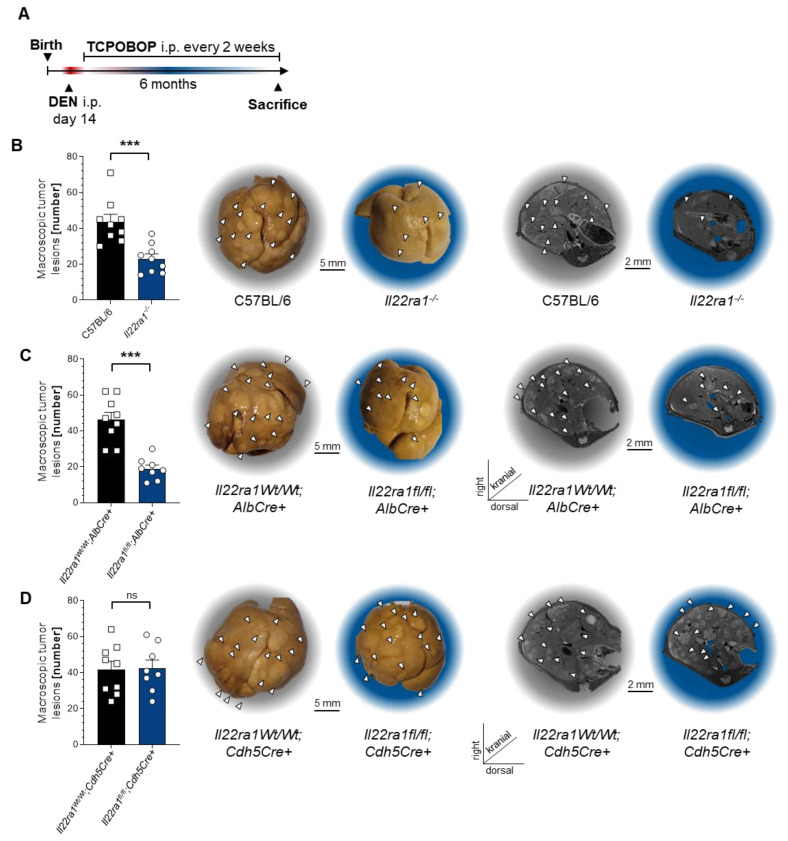
IL-22 signaling in hepatocytes promotes HCC development. (**A**) A schematic timeline describing chemical HCC induction with DEN and TCPOBOP in mice; (**B**) left: number of macroscopic tumor lesions in livers of 6-month-old mice that were either wild type (C57BL/6, left, black, *n* = 9), or *IL22ra1*-deficient (*IL22ra1^-/-^*, right, blue, *n* = 8); middle: representative macroscopic pictures; right: representative MRI pictures; (**C**) left: number of macroscopic tumor lesions in livers of 6-month-old mice that were either controls (*IL22ra1^+/+^*;*Alb^Cre+^*, left, black, *n* = 9), or deficient for *IL22ra1* on hepatocytes (*IL22ra1^flox/flox^*; *Alb^Cre+^*, right, blue, *n* = 8); middle: representative macroscopic pictures; right: representative MRI pictures; (**D**) left: number of macroscopic tumor lesions in livers of 6-month-old mice that were either controls (*IL22ra1^+/+^*;*Cdh5^Cre+^*, left, black, *n* = 9), or deficient for *IL22ra1* on endothelial cells (*IL22ra1^flox/flox^*; *Cdh5^Cre+^*, right, blue, *n* = 8); middle: representative macroscopic pictures; right: representative MRI pictures. Data are pooled from 2 independent experiments. Data presented as mean ± SEM. ns: *p* > 0.05; ***: *p* ≤ 0.001 as assessed by Mann–Whitney U test.

**Figure 5 cancers-14-06019-f005:**
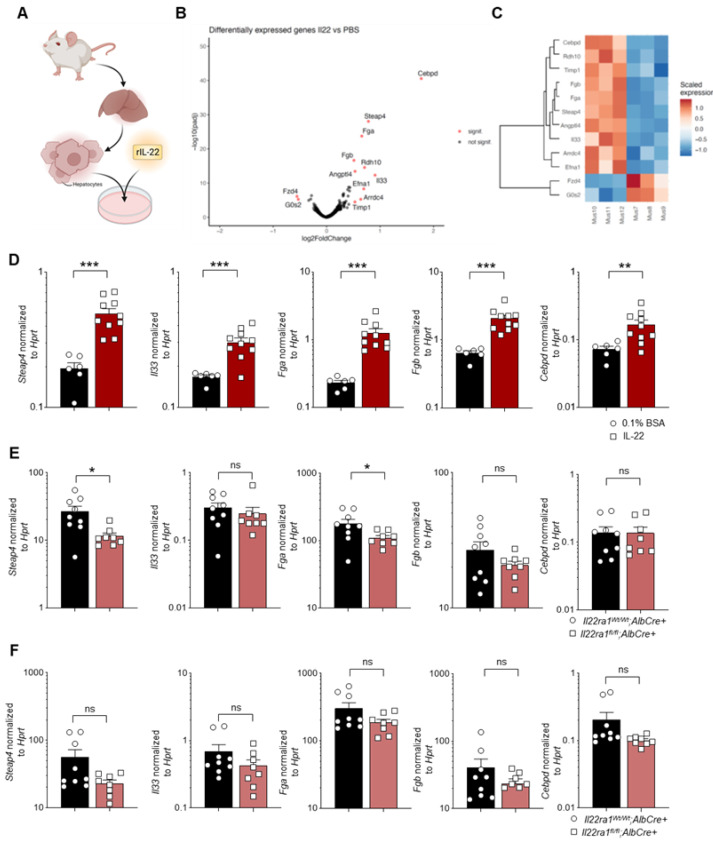
IL-22 signaling induces transcriptional changes in murine hepatocytes. (**A**) A schematic picture describing the experimental setup; (**B**) volcano-plot depicting the differently expressed genes in hepatocytes upon in vitro PBS-stimulation (left, *n* = 3) versus rIL-22 stimulation (right, *n* = 3); (**C**) heatmap and hierarchical clustering of differentially expressed genes in hepatocytes upon in vitro IL-22 stimulation (left, *n* = 3) versus PBS-stimulation (right, *n* = 3); (**D**) relative expression of *STEAP4*, *IL33*, *FGA*, *FGB* and *CEBPD* in comparison to *HPRT* in murine hepatocytes that were either stimulated with 0.1% BSA as control (black, *n* = 6) or rIL-22 (red, *n* = 10); (**E**) relative expression of *STEAP4*, *IL33*, *FGA*, *FGB* and *CEBPD* in comparison to *HPRT* in the livers of *IL22ra1^+/+^*; *Alb^Cre+^* (black, *n* = 9) or *IL22ra1^flox/flox^*; *Alb^Cre+^* (red, *n* = 8) 6-month-old mice that underwent chemical HCC induction as outlined above; (**F**) relative expression of *STEAP4*, *IL33*, *FGA*, *FGB* and *CEBPD* in comparison to *HPRT* in the livers of *IL22ra1^+/+^*;*Cdh5^Cre+^* (black, *n* = 9) or *IL22ra1^flox/flox^*; *Cdh5^Cre+^* (red, *n* = 8) mice 6 months after chemical HCC induction. Data are pooled from 2 independent experiments. Data presented as mean ± SEM. ns: *p* > 0.05; *: *p* < 0.05; **: *p* ≤ 0.01; ***: *p* ≤ 0.001 as assessed by Mann–Whitney U test.

**Figure 6 cancers-14-06019-f006:**
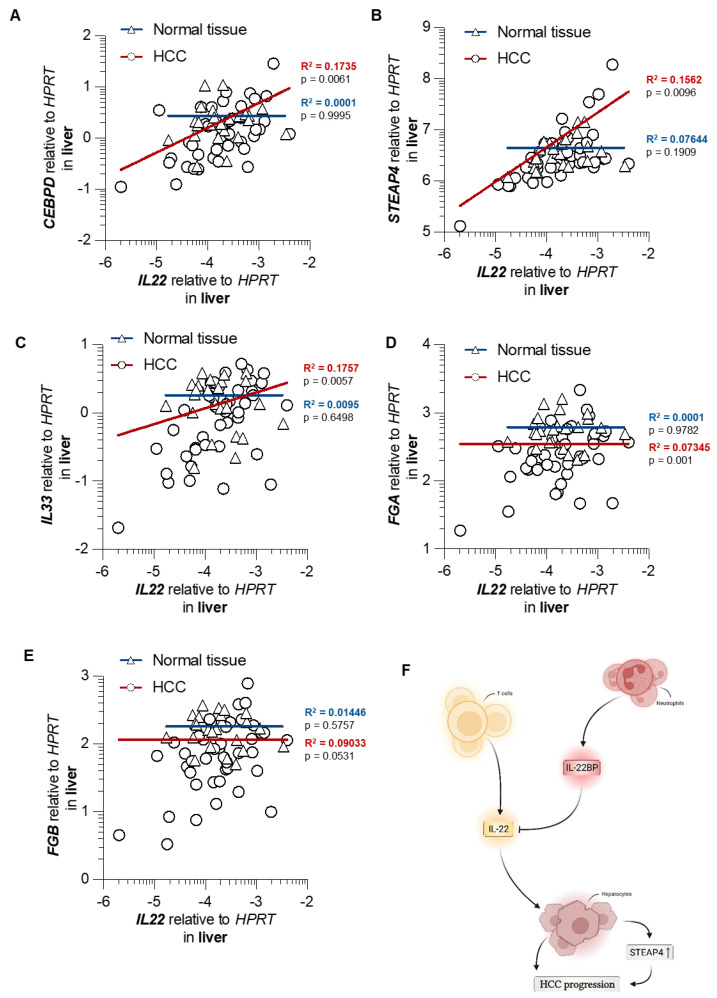
*IL22* expression is associated with the expression of *CEBPD*, *STEAP4*, and *IL33* in HCC but not in healthy livers. (**A**) correlation of relative expression of *IL22* and *CEBPD* in comparison to *HPRT* in liver samples with HCC (red, *n* = 42) and in normal liver tissue (blue, *n* = 24); (**B**) correlation of relative expression of *IL22* and *STEAP4* in comparison to *HPRT* in liver samples with HCC (red, *n* = 42) and in normal liver tissue (blue, *n* = 24); (**C**) correlation of relative expression of *IL22* and *IL33* in comparison to *HPRT* in liver samples with HCC (red, *n* = 42) and in normal liver tissue (blue, *n* = 24); (**D**) correlation of relative expression of *IL22* and *FGA* in comparison to *HPRT* in liver samples with HCC (red, *n* = 42) and in normal liver tissue (blue, *n* = 24). (**E**) correlation of relative expression of *IL22* and *FGB* in comparison to *HPRT* in liver samples with HCC (red, *n* = 42) and in normal liver tissue (blue, *n* = 24). Displayed are all samples with detectable *IL22* expression from 60 samples analyzed of HCC and 37 samples analyzed of healthy livers. Each data point represents one patient sample. (**F**) graphical representation depicting the role of the IL-22−IL-22BP axis in HCC.

## Data Availability

Data are included within the article or its Appendix A. RNA sequencing data will be provided upon reasonable request.

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
