# Peer review of "A Critical Role of the IL-22–IL-22 Binding Protein Axis in Hepatocellular Carcinoma"

_cancers, 2022, doi:10.3390/cancers14246019_

Round 1

Reviewer 1 Report

The present study investigated the pathological role of IL-22 signaling and its natural antagonist IL-22 binding protein in hepatocellular carcinoma (HCC). The results showed that IL-22BP is protective and abrogation of IL-22 signaling was associated with reduced STEAP4 expression. The data is interesting and of clinical implications. Some minor points are listed as below.

1. In vitro assays may help delineate the effect of IL-22 signaling on the STEAP4 expression.

2. Statistical significance seems to be missing in Figure 3D and E.

Reviewer 2 Report

This is an elegant work that reveals the critical role of the IL-22-IL22BP axis in hepatocellular carcinoma and provides the foundation for developing new therapeutical approaches in HCC. I have a small suggestion as follow:

The author indicated that IL-22BP was abundantly secreted by neutrophils in Figure3E, I think if the author can add the significant statistics assay in Figure3E will be better.

Reviewer 3 Report

This manuscript addresses important questions in this field and can provide potentially useful information. (The delicate balance between IL-22 and IL-22BP generally plays a key role in the physiological symphony of signaling for the expression of proteins and subsequently regulatory networks).

You have obtained good data, but you can do better in discussing and understanding the concept.
The following are suggested to be done if possible:
- It is better to display the heatmap as a cluster.
- I miss a more detailed analysis of the heatmap data of mRNA expression that the cell line treated with IL-22BP or double treatment of IL-22 and IL-22BP.                                                                                                               - It is interesting to know the Protein interaction network for mRNA expression (treatment of IL-22 and IL-22BP).                                                     - Schematic illustration to express the mechanism between IL-22 and IL-22BP.
